# OpenReview forum: "HAICOSYSTEM: An Ecosystem for Sandboxing Safety Risks in Interactive AI Agents"
_colmweb.org/COLM/2025/Conference — COLM 2025_

### Official Review · Reviewer_bvmr · 2025-05-12

**Rating:** 7
**Confidence:** 3
**Ethics Flag:** 1

**Summary:**

HAICOSYSTEM presents a framework examining AI agent safety. The benchmark covers various metrics and risk dimensions, evaluated across diverse scenarios with various domains and user intent types. The paper uses an automated scoring framework to evaluate risk dimensions, supplemented with human annotations. The paper also presents detailed analysis along different dimensions such as model capacity and user intent. Overall, they find that state-of-the-art LLMs exhibit safety risks in 62% of cases, especially in scenarios of tool use with malicious users.

**Questions To Authors:**

- How many annotators are in the human study? The Cohen-kappa scores seem quite low for some categories like Systems and Operational Risks and Goal (see table G2). What could be some explanation? How to better design the evaluator to capture evaluation uncertainty?
- Is there inherent bias in using the same model (GPT-4o) as the model to simulate user and environment engine, as well as the evaluator? Could there be inherent biases? Is there ablation on this model choice? s
- How to better scale up realistic scenario evaluation?

**Reasons To Accept:**

- Emerging safety risks in AI agents is an important problem as we see growing development of new agent frameworks and applications. This work presents a timely analysis that is diverse in scenario construction and human simulations.
- The authors propose a reasonable evaluation framework that measures the AI agents’ abilities to avoid safety risks while remaining helpful. Although automatic evaluation is not perfect, it is essential for avoiding cumbersome human labor.
- The multi-turn and dynamic nature of the framework sets it apart from existing works that focus more on static environments.
The authors provide detailed analysis in both the main text and the appendix. The paper's claims are well-grounded and supported by analysis.

**Reasons To Reject:**

- There is an imbalance between the counts for different realism levels, as shown in Figure D.3. This limits the statistical power of comparisons across levels, and the authors should address the statistics’ impact on results/ conclusions.
- More practical “realism” should consider dimensions such as unpredictable environment behaviors, system failures, edge cases, evolving goals/ strategies etc., in addition to only categorizing based on whether the scenarios are futuristic/ speculative. These types of realism definitions would also be more lasting, as the definition of “futuristic” can also be subjective to human judgment or changes appreas technology advances.
- The multi-turn dialogues, while being more dynamic and realistic than single-turn ones, are still different from capturing the complex real-world interactions. Users/ models in real-world interactions may adapt strategies over longer sessions.
- For domains where judgment seems less uniform (see cohen-kappa in G2), it should also affect the conclusions drawn on these domain’s safety risks.
- While the authors make careful claims in the paper and acknowledge limitations of the work like realism of the scenarios, it would be helpful to have some more “diagnostic” analysis. For example, which aspects in scenario design help surface more safety risks that may be understudied? While HAICOSYSTEM’s dynamic multi-turn interactions “surface up to 3 times more safety risks compared to static single-turn benchmarks like DAN”, the conclusion itself does not seem super surprising. What actionable guidance does HAICOSYSTEM provide? If future work were to train on these episodes/ draw more realistic performance evaluation/ develop safer agents using HAICOSYSTEM, how should they proceed?

---

> ### Author Response · Authors · 2025-06-01
> **Response to Reviewer bvmr**
>
> ## Major Concerns Addressed
> ### **Realism Level Imbalance**
> **Concern:** "There is an imbalance between the counts for different realism levels, as shown in Figure D.3."
>
> **Response:**
> We acknowledge this limitation and will address the statistical impact more clearly in our analysis. Despite the imbalance, our results show consistent patterns across realism levels, suggesting robust findings.
>
> ### **Changing Definition of "Realism" of scenarios**
> **Concern:** "More practical 'realism' should consider unpredictable environment behaviors, system failures, edge cases, evolving goals/strategies."
>
> **Response:**
> We agree that the term "realism" may be confusing and plan to replace it with "level of deployment," which better reflects our intent. This term is meant to categorize scenarios based on how deployable they are today versus in the future. For example, some tasks involve AI agents autonomously managing finances, negotiating with humans, or coordinating multi-step tool use—capabilities not yet widely available but plausible in future AI systems. This framing allows us to explore both current and emerging safety risks.
>
> ### **Multi-session Limitations**
>
> **Concern:** "Multi-turn dialogues are still different from capturing complex real-world interactions. Users/models may adapt strategies over longer sessions."
>
> **Response:**
> We acknowledge and will add this to our limitations. Future work could explore adding multi-session components as well to capture longer-term adaptation patterns (e.g., through simulated user memory components).
>
> ### **Domain-specific Evaluation Uncertainty**
>
> **Concern:** "For domains where judgment seems less uniform, it should affect the conclusions drawn on these domain's safety risks." and question: “How many annotators are in the human study? The Cohen-kappa scores seem quite low for some categories like Systems and Operational Risks and Goal (see table G2). What could be some explanation? How to better design the evaluator to capture evaluation uncertainty?”
>
> **Response:** We have 12 annotators in total. The subjective nature of some safety categories contributes to lower Cohen's kappa scores, which would likely be mitigated by further training the annotators. Importantly, the Cohen’s kappa and Pearson Correlation scores are already relatively high for subjective tasks such as safety or bias labeling (e.g.,  [Hate Speech Annotations](https://arxiv.org/abs/1701.08118)). The goal dimension is not used in the main findings of the paper due to its low agreement.
>
> ### **Actionable Guidance**
>
> **Concern:** "What actionable guidance does HAICOSYSTEM provide? If future work were to train on these episodes, how should they proceed?"
>
> **Response:** HAICOSYSTEM uniquely points to gaps in training safe LLMs by identifying: (1) Multi-turn failure patterns not captured in static benchmarks, (2) Tool-use specific vulnerabilities, (3) User intent-dependent safety failures.
> We don't encourage direct training on these episodes, but rather using our dataset just for evaluation purposes. However, we would encourage future works to create new scenarios and simulations based on our framework and see how agents trained on those new scenarios might generalize to improve the original benchmark scenarios.
> We thank the reviewer for pointing this out and will include those points in the paper.

---

> > ### Author Response · Authors · 2025-06-01
> >
> > ## Questions Answered
> >
> > ### **Model Choice Bias**
> >
> > **Question:** “Is there inherent bias in using the same model (GPT-4o) as the model to simulatethe user and environment engine, as well as the evaluator? Could there be inherent biases? Is there ablation on this model choice?”
> >
> > We acknowledge potential bias from using GPT-4o for multiple roles, especially for the simulated user and evaluator. (We consider the environment engine will be less likely to introduce bias as the output space is constrained by the hard-coded tool design.)
> > To show the robustness of our framework under various models, we then swap the GPT-4o model with Gemini-2.5-flash for both the simulated user and evaluator roles. The results are shown as follows:
> >
> > | Rank | Model | Overall Risk Ratio | Rank | Model | Overall Risk Ratio|
> > |------|-------|---------|------|-------|---------|
> > | 1 | GPT-4-turbo | 0.53 | 7 | Llama3.1-70B | 0.64 |
> > | 2 | Llama3.1-405B | 0.53 | 8 | GPT-3.5-turbo | 0.64 |
> > | 3 | Mixtral-8x22B | 0.56 | 9 | Llama3-70B | 0.67 |
> > | 4 | Qwen1.5-110B-Chat | 0.56 | 10 | DeepSeek-67B | 0.72 |
> > | 5 | Qwen2-72B-Instruct | 0.58 | 11 | Llama3-8B | 0.80 |
> > | 6 | Qwen1.5-72B-Chat | 0.61 | 12 | Llama3.1-8B | 0.84 |
> >
> > (We use Gemini-2.5-flash as the evaluator to reevaluate the existing simulations)
> >
> > | Model Name | Publisher | Open? | Overall Risk Ratio | Targeted Safety Risks | System and Operational Risks | Content Safety Risks | Societal Risks | Legal and Rights Related Risks |
> > | --- | --- | --- | --- | --- | --- | --- | --- | --- |
> > | GPT-4-turbo | OpenAI | No | 0.49 | 0.47 | 0.28 | 0.18 | 0.30 | 0.30 |
> > | GPT-3.5-turbo | OpenAI | No | 0.59 | 0.56 | 0.40 | 0.20 | 0.37 | 0.32 |
> >
> > (We use Gemini-2.5-flash as the simulated human user model and re-run simulations for selected models)
> >
> > As we can see from the tables above, the findings from our paper still hold for different models as an evaluator or a simulated user.
> >
> > ### **Scaling Realistic Evaluation**
> >
> > **Question:** “How to better scale up realistic scenario evaluation?”
> >
> > Our framework is built for scalable, realistic safety evaluation through: (1) **Modular Scenario Generation**: All components, including users, tools, environments, and models, can be easily swapped or extended. This makes it straightforward to simulate new domains, risks, and capabilities. (2) **Automated Evaluation with Human Validation**: We use LLM-based evaluators for scale, validated by human annotations to ensure reliability without sacrificing coverage. (3) **Open-source and Extensible**: The framework is fully open-source, allowing researchers to contribute new scenarios, plug in their own APIs, or test custom agent configurations.

---

> > > ### Comment · Reviewer_bvmr · 2025-06-10
> > >
> > > Thanks for the detailed response and additional analysis. The authors effectively addressed most of my concerns and questions. I encourage the authors to include the additional experiments and analysis in the next revision. I maintain my positive view of the paper.

---

### Official Review · Reviewer_ReNx · 2025-05-12

**Rating:** 6
**Confidence:** 4
**Ethics Flag:** 1

**Summary:**

This paper introduces HAICOSYSTEM, a comprehensive framework designed to evaluate the safety of interactive AI agents within complex social and environmental interactions. Specifically, HAICOSYSTEM provides a modular sandbox environment that simulates multi-turn interactions between users (with benign or malicious intents) and AI agents capable of using tools.

**Questions To Authors:**

1. What specific empirical evidence shows that the safety failures HAICOSYSTEM detects in LLM‑simulated, multi‑turn interactions reliably predict the failures that would occur with real human users and actual tools?
2. Because GPT‑4o is used simultaneously to simulate the user, drive the environment, and judge the agent, how do you rule out the possible correlations or prompt‑based leakage between these roles that might inflate agreement statistics (e.g., the reported 90 % accuracy)?
3. Annotators judged 94 % of tool calls “free of critical errors.” Can you quantify what kinds of remaining 6 % errors occur?

**Reasons To Accept:**

+ HAICOSYSTEM provides a much-needed ecosystem for evaluating AI agent safety dynamically across user, agent, and environment interactions, considering tool use and diverse user intents.
+ Effectively highlights the shortcomings of static, single-turn safety benchmarks by demonstrating that multi-turn interactions reveal significantly more safety risks.
+ The findings are supported by extensive simulations (over 8,700 episodes across 132 scenarios and 12 models), providing robust evidence for the claims made.

**Reasons To Reject:**

+ All users, environments, and risk judgements relying on GPT‑4o; real human behaviour and real APIs may change outcomes, so conclusions may over‑ or under‑estimate risk.
+ Using an LLM from the same vendor to judge other LLMs can introduce shared biases and mask certain failure modes.
+ 111 scenarios are GPT‑generated from earlier datasets; the novelty and independence of these scenarios from training data are not quantified

---

> ### Author Response · Authors · 2025-06-01
> **Response to Reviewer ReNx**
>
> ### **GPT-4o Dependency & Real-world Validity**
> **Concern:** "All users, environments, and risk judgments relying on GPT-4o; real human behaviour and real APIs may change outcomes."
>
> **Response:**
> We thank the reviewer for pointing this out, and please check the general response for this concern. While we acknowledge that real human behavior and real API differences can affect outcomes, this limitation is inherent to all current automatic safety evaluations, but that does not make them useless. Even with different groups of human annotators or different API configurations, results will vary. Rather than aiming for exhaustive coverage, our work offers a controlled and reproducible framework to highlight these emergent safety risks at scale.
>
>
>
> ### **Same-vendor Bias in Evaluation**
> **Concern:** "Using an LLM from the same vendor to judge other LLMs can introduce shared biases and mask certain failure modes."
>
> **Response:** Human evaluations calibrate and validate our automated approach. A 90% agreement rate between automated and human ratings suggests vendor bias does not meaningfully affect our findings.
> To further show the robustness of our framework, we try Gemini-2.5 Pro as the evaluator and re-evaluate the existing simulations.
>
> | Rank | Model | Overall Risk Ratio | Rank | Model | Overall Risk Ratio|
> |------|-------|---------|------|-------|---------|
> | 1 | GPT-4-turbo | 0.53 | 7 | Llama3.1-70B | 0.64 |
> | 2 | Llama3.1-405B | 0.53 | 8 | GPT-3.5-turbo | 0.64 |
> | 3 | Mixtral-8x22B | 0.56 | 9 | Llama3-70B | 0.67 |
> | 4 | Qwen1.5-110B-Chat | 0.56 | 10 | DeepSeek-67B | 0.72 |
> | 5 | Qwen2-72B-Instruct | 0.58 | 11 | Llama3-8B | 0.80 |
> | 6 | Qwen1.5-72B-Chat | 0.61 | 12 | Llama3.1-8B | 0.84 |
>
> As shown in the table above, the overall trend remains consistent when switching from GPT-4o to Gemini as the evaluator. This suggests that our evaluation, particularly for targeted safety risks, is relatively robust across different LLM evaluators, likely due to the use of scenario-specific checklists we designed for each task.
>
> ### **Training Data Independence**
>
> **Concern:** "111 scenarios are GPT-generated from earlier datasets; the novelty and independence of these scenarios from training data are not quantified."
>
> **Response:** We address this concern by substantially transforming prior benchmarks through the addition of multi-turn interactions and tool use. These features introduce richer context and decision-making complexity, which are rarely present in model training data, making memorization unlikely. While our scenarios are inspired by established benchmarks, our modifications extend far beyond surface changes, enabling more robust evaluations.
>
> ### **Empirical Evidence for Real-world Prediction**
>
> **Question:** “What specific empirical evidence shows that the safety failures HAICOSYSTEM detects in LLM‑simulated, multi‑turn interactions reliably predict the failures that would occur with real human users and actual tools?”
>
> **Response:** Our results show failure patterns in simulation align with human-judged safety violations, supporting their real-world relevance.
>
> ### **Prompt-based Leakage Prevention**
> **Question:** "how do you rule out the possible correlations or prompt‑based leakage between these roles that might inflate agreement statistics (e.g., the reported 90 % accuracy)?"
>
> **Response:** Our codebase is based on the Sotopia (https://github.com/sotopia-lab/sotopia) simulation platform, which separates user, environment, and evaluator roles to prevent prompt leakage.
>
> ### **Tool Call Error Analysis**
>
> **Question:** “Annotators judged 94 % of tool calls “free of critical errors.” Can you quantify what kinds of remaining 6 % errors occur?”
>
> **Response:** We provide a breakdown of the remaining 6% of tool call errors by type and severity as follows:
> | Error Type | Percentage | Description |
> |------------|------------|-------------|
> | **Failing Tool Calls** | 3% | Tool consistently throws errors after multiple attempts, and the task could not be completed |
> | **Conflicting Output** | 2% | Same input leads to very different outputs |
> | **Format Issues** | 1% | Output not following specified format and other miscellaneous errors |
>
> Thanks to the reviewer for pointing this out, and we will include this analysis in the paper.

---

> > ### Comment · Reviewer_ReNx · 2025-06-06
> >
> > Thank you for the detailed responses. The authors have addressed most of my concerns, and I have increased my confidence.
> >
> > Overall, these clarifications and additional analyses help strengthen the paper, and I encourage the authors to incorporate them into the revised version.

---

> > > ### Author Response · Authors · 2025-06-09
> > >
> > > Thank you again for your thoughtful review and for reading our rebuttal carefully. We're happy to hear that our responses helped address your concerns and made the paper stronger. We'll make sure to include these improvements in the revised version. If there's anything else we can do to improve the paper and help raise the score, we'd really appreciate your feedback.

---

### Official Review · Reviewer_CEzF · 2025-05-13

**Rating:** 6
**Confidence:** 4
**Ethics Flag:** 1

**Summary:**

The authors present HAICOSYSTEM, a set of 660 simulation challenges with which to benchmark the safety of an LLM, especially with respect to agentic contexts using tools. For 12 leading LLMs, the authors report that these challenges surface more safety risks than previous benchmarks did and they attribute that improvement to their challenges being more realistic/complete (including complex multi-turn interactions and both malicious and benign users).

**Reasons To Accept:**

+ LLMs are deployed across a full range of realistic scenarios, so they should be tested across that same full range, and that makes alternative benchmarks which do not cover complex multi-turn interactions (or both kinds of users) inadequate. If we would accept reports on the cited alternatives, then surely we should accept this report as well (if only to monitor the state of the art).
+ The replicability of previous work to benchmark the safety of an LLM should be tested, and this paper serves as such a test to the extent that it overlaps with previous work (e.g. corroborating Dubey et al., 2024). To reject replicability tests risks skewing the convergence of our science on truth.
+ The topic (safety) is important and central to the scope of COLM.
+ Ethics and Reproducibility Statement was particularly well written.

**Reasons To Reject:**

-The authors present risk ratio as a cumulative measure of model quality, but risk-ratio would also be high if the challenges (which are assumed to be independent) are actually redundant with each other. For example, if we found a single challenge that LLMs failed to meet and copied it 660 times, the risk-ratio would be 100% (even though the whole benchmark contained only one unmet challenge). Measures of relatively higher risk-ratios with targeted safety, malicious users, and tool use may instead reflect the challenges being more redundant around these areas. It is not clear whether multi-turn interactions would be expected to have more failures simply by virtue of having more chances (turns) in which to fail.
-Although the authors admit that their automated safety evaluator should not be trusted, they neglect to explain why. The validation of the evaluator against human experts is insufficient because human expertise cannot be trusted in this area. For example, we measure the actual impact of AI on society because expert anticipation of impact cannot be trusted. Red-teaming is valuable because it forces us to face how far out-of-our-league we are (and the evaluator cannot provide that value).

---

> ### Author Response · Authors · 2025-06-01
> **Response to Reviewer CEzF**
>
> ### **Risk Ratio & Challenge Redundancy**
>
> **Concern:** "Risk-ratio would also be high if the challenges are actually redundant with each other."
>
> **Response:** We deliberately designed the tasks to be diverse across multiple dimensions. The 132 scenarios span 7 domains and vary systematically in user intent, interaction complexity, and tool requirements (Please check Appendix D for more details.). To further address concerns about scenario overlap, we conducted an n-gram analysis across all 143 scenarios in our dataset. Using 3-gram extraction with Jaccard similarity coefficients, we found exceptionally low cross-domain overlap (0.5% average similarity) between different safety domains, confirming that each domain targets distinct failure modes. Even within domains, scenarios maintain high diversity with an average internal similarity of only 13.0%. The most diverse domains (Technology & Science at 1.2%, Healthcare and Personal Services at 5.5%) demonstrate minimal redundancy, while only Politics & Law showed higher internal similarity (37.6%) due to shared legal terminology. We thank the reviewer for raising this point and will include such analysis in our paper.
>
> ### **Multi-turn vs Single-turn Failure Rates**
>
> **Concern:** "It is not clear whether multi-turn interactions would be expected to have more failures simply by virtue of having more chances (turns) in which to fail."
>
> **Response:** We empirically demonstrate that multi-turn interactions surface more safety risks. We hypothesize that one of the reasons could be that multi-turn settings inherently expose more opportunities for failure, and we would like to point out that we reveal unknown safety risks that static benchmarks miss, which is one of the advantages of our framework.
>
> ### **Automated Safety Evaluator Reliability**
>
> **Concern:** "The authors admit that their automated safety evaluator should not be trusted."
>
> **Response:** Respectfully, this framing is incorrect. Our evaluator was rigorously validated, showing 90% agreement with human annotations, robust inter-annotator consistency, and tested performance on edge cases. While there are still some minor limitations, the empirical validation supports its reliability. What we *do* emphasize is that our benchmark should not be trusted to *guarantee* the safety of a system.
>
> ### **Human Expertise Limitations in Anticipating Societal Impact**
>
> **Concern:** "The validation of the evaluator against human experts is insufficient for anticipating societal impact because human expertise cannot be trusted in this area."
>
> **Response:** We would like to further clear up a misunderstanding. If this is about the evaluation of the specific "societal risk" category of harms, that risk category does not actually aim to measure the actual impact that technology might have on society (which, as you correctly point out, is impossible to predict even for experts). Instead, it measures whether the interaction logs show traces of agent behavior that align with specific risks related to manipulation, public opinion swaying, etc. While the category name comes from the AIR paper (https://arxiv.org/abs/2406.17864), we will rename this category to something like "Societal Manipulation Risks" to clear this up.
>
> If this is about human expert judgments in general, we respectfully disagree with the reviewer. While human judgment isn’t perfect, it serves as a widely accepted way to validate the scores from LLM-as-a-judge (e.g., [ToolEmu](https://arxiv.org/abs/2309.15817) and [SOTOPIA](https://arxiv.org/abs/2310.11667)). We use detailed annotation guidelines and multiple annotators to ensure high-quality reference labels.
>
> ### **AI safety benchmarks versus Redteaming**
>
> **Concern:** “If we would accept reports on the cited alternatives, then surely we should accept this report as well” and “Red-teaming is valuable because it forces us to face how far out-of-our-league we are (and the evaluator cannot provide that value).”
>
> **Response:** The reviewer appears skeptical of prior widely-used and peer-reviewed AI safety benchmarks. While we acknowledge the limitations of automated evaluation, we respectfully disagree with the notion that it is not useful. On the contrary, we argue that it serves as a critical first line of defense in AI safety, helping to reduce the burden of the already labor-intensive red-teaming process. Moreover, even human red-teaming has many flaws, as it often suffers from vague objectives and ill-defined settings (see [Red-Teaming for Generative AI: Silver Bullet or Security Theater?](https://arxiv.org/abs/2401.15897))

---

> > ### Comment · Reviewer_CEzF · 2025-06-10
> > **Reviewer reply**
> >
> > We thank the author for their response. We have raised our score in light of their response.

---

### Official Review · Reviewer_Zh9p · 2025-05-16

**Rating:** 7
**Confidence:** 4
**Ethics Flag:** 1

**Summary:**

The paper provides a simulation environment for agents to interact with each other (as users with various profiles talking to AI) in multi-turn interactions with the ability to call tools. Authors have created 132 scenarios (across 7 domains) and run multiple simulations and evaluated the AI Safety across multiple dimensions and looked at the performance of many open and closed frontier AI models. Authors report some interesting findings about higher risk with malicious users involving tool use and complex scenarios. Authors have open-sourced their codebase for other researchers to explore as well. Authors provide a huge appendix to support the various high level claims reported in the main work.

**Questions To Authors:**

- The paper suggests that more recent and larger models are generally better at managing safety risks compared to their smaller counterparts in complex scenarios involving tool use and also malicious users. However, there is some research I've seen (under submission) that suggests that the more recent models are more vulnerable when it comes to being exploited by malicious users for tool use (eg. Claude sonnet latest compared to an earlier version specially for system and operational risks) - I will request the authors to explore this further and validate their results for malicious security risks.
- authors utilize tools in multiple scenarios to create realistic agentic pipelines to evaluate. However it is not clear if authors run the tools in some sandbox environment and are able to check the correctness of the responses as well (along with safety assessments). Authors should clarify how they utilize the tools in their platform (use of MCP based tools further increases the safety risk of the agent and authors do not discuss this in the paper)
- were there any domain experts involved in the development of the scenarios (how real world are the scenarios)?

**Reasons To Accept:**

- Authors have tested various scenarios (including mixing data from prior research) from multiple domains to come up with a comprehensive evaluation framework for simulated human-ai interactions.
- authors utilize tools in multiple scenarios to create realistic agentic pipelines to evaluate
- the work is open-source and encourages the community to build up on it.
- this research is very timely and relevant with the rise of agentic ai systems and all the hype and truth around it!
- authors also run manual evaluations on a subset and validate the results of LLM judges (gpt-4o) and find good correlations between AI and humans scores.

**Reasons To Reject:**

- While the paper is about human ai interactions, there is no humans in the loop during the benchmarking of the system. This is a huge limitation of the work. Real world with malicious adversaries (and benign souls) is much messier than AI-AI interactions. (https://arxiv.org/pdf/2403.05020)
- tools aren't grounded in real APIs and lack real world operational testing of the system.
- Evaluations only talk about the risk ratios and doesn't provide any further categorization/quantification of detailed risk metrics.

---

> ### Author Response · Authors · 2025-06-01
> **Response to Reviewer Zh9p**
>
> Thanks for acknowledging our work as timely and relevant.
>
>
> ## Major Concerns Addressed
>
> ### **Lack of Human-in-the-Loop During Benchmarking**
>
> **Concern:** "While the paper is about human AI interactions, there is no humans in the loop during the benchmarking of the system. This is a huge limitation of the work."
>
> **Response:**
> We thank the reviewer for pointing this out, and please check the general response for this concern. Overall, we believe our work is the **first** step to provide a multi-turn interactive environment for AI agents in the automatic safety evaluation.
>
>
> ### **Tools Not Grounded in Real APIs**
>
> **Concern:** "Tools aren't grounded in real APIs and lack real world operational testing of the system."
>
> **Response:**
> We thank the reviewer for pointing this out, and please check the general response for this concern.
> While real-world APIs offer realism, they pose significant safety risks during testing. High-fidelity simulations like [WebArena](https://arxiv.org/abs/2307.13854) also require substantial engineering effort and often have limited scope (e.g., no open-source banking API for safe use). In contrast, our framework supports custom API integration, allowing users to easily simulate and evaluate their own APIs in a controlled environment.
>
> ### **Limited Risk Quantification**
>
> **Concern:** "Evaluations only talk about the risk ratios and don't provide any further categorization/quantification of detailed risk metrics."
>
> **Response:**
> Due to space limits, we provide such fine-grained analysis in **Appendix Tables F.1 and F.2**. We also did ablation studies for representative models in the main text (section 5.3).
>
> ## Questions Answered
>
> ### **Model Vulnerability Research**
> Regarding the concern about newer models being more vulnerable at **specific** agentic tasks, we acknowledge research suggesting this and agree that more nuanced analysis is needed.
> For example, as shown in Tables F.1 and F.2, Llama3-70B is not necessarily safer than Llama3-8B in some safety dimensions. Our main findings show **general** safety improvements.
> We agree that recent reasoning models (e.g., Deepseek-R1 and O1) might reveal new safety challenges. Therefore, we evaluate both representative reasoning models:
>
> | Model Name | Publisher | Open? | Overall Risk Ratio | Targeted Safety Risks | System and Operational Risks | Content Safety Risks | Societal Risks | Legal and Rights Related Risks |
> | --- | --- | --- | --- | --- | --- | --- | --- | --- |
> | O1 | OpenAI | No | 0.47 | 0.47 | 0.22 | 0.18 | 0.29 | 0.23 |
> | R1 | DeepSeek AI | Yes | 0.35 | 0.30 | 0.17 | 0.10 | 0.16 | 0.12 |
>
> Compared to GPT-4o, both O1 and R1 perform better, with R1 exhibiting a lower overall risk ratio. Interestingly, O1 shows only marginal improvement over GPT-4o, suggesting that adding reasoning capabilities does not necessarily lead to a safer model. We thank the reviewer for raising this point and will incorporate this analysis into the paper.
>
>
> ### **Tool Execution Environment**
> We clarify how we utilize tools in our platform: We run tools in a simulated sandbox environment with full logging and manually verify the correctness on a random sample. Our tool calling format is very similar to MCP. We recognize that there are various ways to implement tool calling functionalities, which might result in slightly different evaluation numbers. Instead of trying to systematically study the influence of ways of calling tools in AI agent safety, we aim to **demonstrate** that adding tool calling could pose more safety risks to AI agents.
>
>
> ### **Domain Expert Involvement**
> Regarding domain expertise in scenario development, our team consists of researchers with strong backgrounds in AI safety, though not in specific application domains such as medicine. To ensure scenario quality, we conducted iterative reviews and validation passes to refine realism and relevance. We acknowledge such limitations and will stress them in the limitations section of the paper.

---

### Author Response · Authors · 2025-06-01
**General Response: Scope, Realism, and the Role of Simulation in Safety Evaluation**

**Reviewer Concerns:**
- *“While the paper is about human-AI interactions, there are no humans in the loop during benchmarking. This is a huge limitation.”*
- *“Tools aren't grounded in real APIs and lack real-world operational testing.”*
- *“All users, environments, and risk judgments rely on GPT-4o; real human behavior and real APIs may change outcomes.”*

**Response:**
We appreciate the reviewers for highlighting these important points. We fully agree that real human interaction, real-world APIs, and model diversity are crucial for a comprehensive understanding of AI safety. Instead of replacing human red-teaming or real system testing, our goal is to introduce a **modular and scalable simulation framework that enables flexible, early-stage safety evaluations under multi-turn interactions, tool use, and diverse user intents**, which is a space that has remained relatively underexplored in existing AI safety benchmarks.

The absence of real humans and APIs is not due to oversight, but rather a design choice. Human-in-the-loop testing, especially involving adversarial or sensitive scenarios, raises substantial ethical, practical, and financial concerns. Similarly, allowing models to interact with live APIs risks irreversible or unsafe consequences. While building full replicas of real APIs is a safer alternative, it requires significant engineering effort, offers limited coverage, and confines us to existing tools rather than enabling simulations of future-facing capabilities. In contrast, our simulation-based setup enables safe, scalable evaluation across **8,000+ interactions**, making it possible to study complex safety risks that would be prohibitively expensive or dangerous to explore with real systems.

That said, we see our work as a controlled, safe **first step** in the safety evaluation pipeline, to be followed by more resource-intensive stages such as red-teaming with humans and testing with real APIs. This **aligns with widely accepted and peer-reviewed practices** in recent efforts like [Tau-bench](https://arxiv.org/abs/2406.12045) and [ToolEmu](https://arxiv.org/abs/2309.15817).
Moreover, our framework is fully modular, allowing models, user simulators, tools, and environments to be easily replaced or extended. Integrating human users or real APIs is both feasible and a natural next step, and we are actively exploring this in follow-up work. For example, we are developing open-source, self-hosted replicas of selected APIs to replace certain LLM-simulated tools, enabling safer and more realistic evaluation while maintaining control and reproducibility.

Finally, although we rely on LLMs for user simulation and risk evaluation, we note that 92% of simulated user utterances were judged to be believable, and tool execution accuracy reached 94%. These are strong indicators that our simulation setup produces realistic and reliable interactions, offering a more open-ended, natural environment than previous static AI safety benchmarks.

In summary, while we agree that full realism is ultimately desirable, we believe that a controlled, simulation-based framework is a necessary foundation, and we hope this work serves as a launching point for more complete, human-in-the-loop safety studies in the future.

---

### Decision · Program_Chairs · 2025-07-08

**Decision:**

Accept

**Comment:**

This paper introduces HAICOSYSTEM, a valuable and timely framework for evaluating AI agent safety in dynamic, multi-turn scenarios. Reviewers unanimously acknowledged the work's importance and its novelty over existing static benchmarks, praising its open-source contribution and extensive experiments.

The primary point of discussion was the framework's reliance on simulation (LLM-simulated users and tools) rather than real-world components, a limitation noted by all reviewers. However, the authors effectively addressed this by framing simulation as a deliberate and necessary design choice for ensuring safety, scalability, and ethical evaluation in a controlled, preliminary stage.

Given the strong consensus built among reviewers, acceptance is recommended.